# Climate Change Risk Assessment for Kurunegala, Sri Lanka: Water and Heat Waves

**Hanna Cho**

Korea Adaptation Center for Climate Change, Korea Environment Institute, 232 Gareum-ro, Sejong 30121, Korea; hncho@kei.re.kr

**Abstract:** Sri Lanka is experiencing various social and environmental challenges, including drought, storms, floods, and landslides, due to climate change. One of Sri Lanka's biggest cities, Kurunegala, is a densely populated city that is gradually turning into an economic revitalization area. This fast-growing city needs to establish an integrated urban plan that takes into account the risks of climate change. Thus, a climate change risk assessment was conducted for both the water and heat wave risks via discussions with key stakeholders. The risk assessment was conducted as a survey based on expert assessment of local conditions, with awareness surveys taken by residents, especially women. The assessment determined that the lack of drinking water was the biggest issue, a problem that has become more serious due to recent droughts caused by climate change and insufficient water management. In addition, the outbreak of diseases caused by heat waves was identified as a serious concern. Risk assessment is integral to developing an action plan for minimizing the damage from climate change. It is necessary to support education and awareness in developing countries so that they can perform risk assessment well and develop both problem-solving and policy-making abilities to adapt to a changing climate.

**Keywords:** climate change adaptation; climate change risk assessment; climate resilience; Kurunegala; risk factors of water; heat waves

## 1. Introduction

Sri Lanka is an island located in the Indian Ocean (Figure 1), which is affected by various natural hazards, including weather-related events, such as cyclones, monsoonal rain and subsequent flooding, and landslides [1]. Sri Lanka has experienced significant and systematic atmosphere warming in all regions [2]. Sri Lanka was one of the countries most affected by climate change in 2018 [3]. The year 2018 began with severe monsoon rains from 20 to 26 May, affecting 20 districts, resulting in at least 24 deaths, more than 170,000 people affected, and nearly 6000 people displaced [4].

Sri Lanka has a total area of 65,610 km$^2$ and a population of 21.8 million, including 3.704 million urban residents living in 64 municipal areas [5]. It is a lower-middle-income country, with a GDP per capita of USD 3853 (2019). Following 30 years of civil war that ended in 2009, the economy grew at an average of 5.3% from 2010–2019 [6]. The country's propitious location favored rapid development projects, such as mega cities, airports, harbors, urban beautification projects, and major highways, during the last ten years [7].

Kurunegala, located in the north-western province, was once the capital of one of the ancient Kingdoms of Sri Lanka. The city is now the capital of the Kurunegala District and is experiencing rapid growth and development [5], with its main economic sectors based on service and industry [8]. This city is one of the most intensively developed economic and administrative capitals in the north-western Province of Sri Lanka, and it is one of the central cities directly connected to a number of other major capital cities and towns.

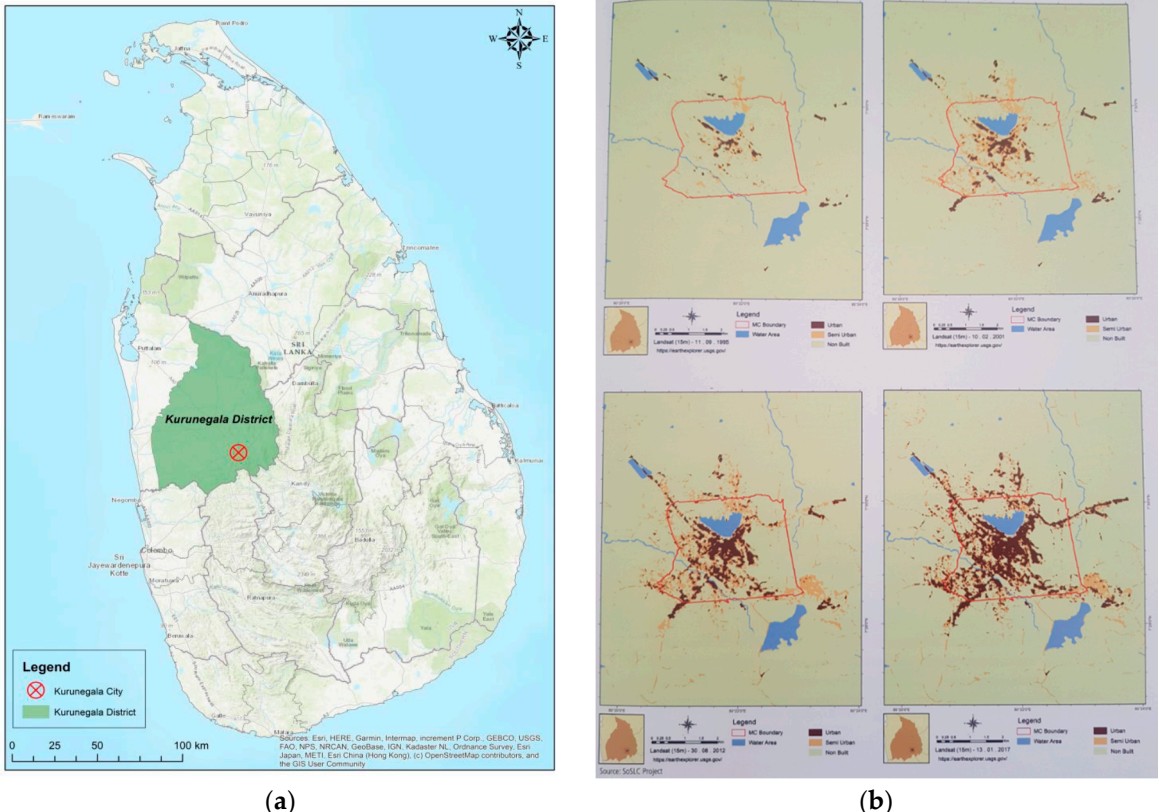

**Figure 1.** Location of Kurunegala (**a**) and the expansion in Kurunegala city (**b**) (1995–2017) (The state of Sri Lankan Cities, 2018).

Most urban systems in Kurunegala are vulnerable to the impacts of climate change and the city has faced challenges while attempting to adapt to climate change. There is an increased need to build with increased risk resilience and develop more and diverse adaptation measures to protect the economic and social wellbeing of city dwellers. To develop a climate change adaptation plan, several processes are needed, i.e., analysis of climate change impacts, vulnerability assessments, and risk assessments are all key processes needed to develop this plan [9–16]. Assessment and analysis of current and future climate change risks and vulnerabilities are required for the development of robust climate change adaptation policies. They inform policy decision-makers of the potential effects of climate change, such that their results can be used to choose among available strategies and their associated technical and social plans for climate change adaptation.

Therefore, this study aims to conduct a climate change risk assessment, the results of which will be used for the development of a climate change adaptation plan for Kurunegala City. This process can also improve the climate resilience of other local cities that are vulnerable to climate change.

## 2. Materials and Methods

Kurunegala city is a densely built residential, business, and commercial area, with a constantly increasing traffic volume and a significantly decreasing amount of green space. Kurunegala city is one of the sprawling cities [5]. Urban sprawl is typically defined as unplanned and uncoordinated low density expansion and involves rapid land consumption as rural spaces transition to urban land use [17]. Urban sprawl can be observed in the Kurunegala city, as demonstrated by satellite images from 1995, 2001, 2012, 2017 (Figure 1). In 1995, the image shows a very small urban space, concentrated in a few areas at the center of the municipal council area. By 2007, we can see significant sprawling expansion in urban and semi-urban areas.

The city's climate remains hot throughout the year and is exacerbated by the rock outcrops surrounding the city, which retain heat during the day. Through the data of the past 10 years, the highest mean monthly temperature was observed during the month of May (26.82 °C), whereas relatively-cooler temperature was experienced in January (23.45 °C). Similar to the nation-wide trend, air temperature in Kurunegala exhibited a general increase. In the year 2016, the mean annual temperature in Kurunegala was 25.92 °C, which is 0.08 to 1.25 °C higher as compared with the previous year [18]. Monsoons occur from May to August and again from October to January. The average annual rainfall in Kurunegala is approximately 2000 mm [2,5]. Sri Lanka experienced a nation-wide drought from the beginning of the year 2016 up to the year 2017. Increase in temperature and lack of rainfall resulted in massive damage in the agricultural produce of the country, especially in the north-western region. Only 458 mm of cumulative rainfall was experienced in Kurunegala in the year 2016. This value was almost 45% lower as compared with the decadal average rainfall experienced in Kurunegala [18]. The city is facing socio-spatial climate vulnerabilities, with the most crucial being extreme heat conditions, periods of flooding, disaster-related public health crises, a decrease in drinking water reserves, and groundwater pollution [9–12].

Climate change risk assessments are conducted to understand how current and projected risk factors impact a community and are a key element of climate change adaptation policy development. They inform policy decision-makers of the potential risks of climate change and provide them with means to evaluate its impacts, as well as to compare different strategies and policies.

For the assessment of climate change risk and vulnerability at the community level, various methods have been studied and performed [10–16,19–22]. In South Korea, both quantitative and qualitative data, as well as information to characterize socio-ecological systems, are required, as both current and predicted risks and vulnerabilities caused by climate change should be included in the assessment [10–12,15,23]. Collecting enough data to understand the impact of climate change in Kurunegala city is difficult; therefore, the assessment focuses on the qualitative method. Quantitative assessment using scientific physical parameters is also needed to increase the accuracy of risk identification in the future.

The qualitative method is a survey-based assessment approach and includes several steps to solve the problem. The purpose of the survey-based assessment approach is to prioritize the risk factors that present the latest problems in Kurunegala city.

The risk assessment was carried out via the following steps.

### 2.1. Step 1: Establish the Context

This stage determines the subject area and purpose. The target area for risk assessment is Kurunegala city, where the risk assessment is limited to water and heat wave problems and is based on key stakeholder information and field observations.

### 2.2. Step 2: Identify the Risks

This step confirms what the types of risks are. To identify the most serious risk in terms of water management and heat stress in Kurunegala city, a detailed set of indicators should be established. Based on Korean examples [23–29], the Kurunegala Municipal Council (KMC) developed their own indicators considering the current situation within their city. KMC developed the following indicators. The difference in Korea's case is that there are no DWR and SDF indicators there. The discovery of indicators for drinking water and sanitation is solely a reflection of the situation in Kurunegala city, Sri Lanka.

1. Drinking water resources risk/vulnerability to drought (DWR); Risk or vulnerability on sources of useful or potentially useful portable water

2. Water management risk/vulnerability (WM); Risk or vulnerability on water management that is the control and movement of water resources to minimize damage to life and property and to maximize efficient, beneficial use

3. Water quality and aquatic ecosystems (WQAE); Risk or vulnerability on water quality and the condition or health of waterways, like rivers, wetlands

4. Water resources risk/vulnerability (WR); Risk or vulnerability on resource of water that is useful or potentially useful, for agricultural, industrial, household, recreational and environmental activities

5. Sanitation risk/vulnerability to droughts and floods (SDF); Risk or vulnerability on sanitation, which is the process of keeping places clean and healthy, especially by providing a sewage system and a clean water supply, due to drought and flood

6. Health risk/vulnerability to floods (HF); Risk or vulnerability of health that is impacted by flood

7. Health and infrastructure risk/vulnerability to heat stress (HIH); Risk or vulnerability on health and infrastructure that is impacted by heat stress and drought

### 2.3. Step 3: Analyze the Risks

Detailed risk factors for each indicator should be analyzed to prioritize the risks of climate change in Kurunegala city. The 84 risk factors determined to be indicators were collected from the Korean national and municipal climate change assessment. KMC reviewed the list of risk factors and revised it to suit their city. Experts and environment ministry officials in the Kurunegala modified and excavated indicators reflecting the characteristics of the city. A total of 57 risk factors were finalized for Kurunegala city (Appendix A).

### 2.4. Step 4: Evaluate the Risks

For the planning, an adaptation action plan for Kurunegala city, we needed to prioritize the risks and find the largest and most serious climate change impacts. Basically, it should be evaluated using objective and quantitative data and information. If quantitative assessment is difficult, it can be diagnosed and predict regional risks through qualitative methods of collecting opinions and discussing them at T/F meetings, expert advice, workshops, etc., using data collected through statistical data, interviews with officials, and surveys. Therefore, a survey was conducted to evaluate the priority of these risk factors because of the lack of quantitative data.

Although it is difficult to quantitatively express the risks of climate change, this study attempted to evaluate climate change with as much objectivity as possible to prepare for its effects. According to the Intergovernmental Panel on Climate Change 5th Assessment Report (IPCC AR5), risk is correlated with "Vulnerability", "Exposure", and "Hazard". Risk is often represented as the probability of an occurrence of hazardous events multiplied by the impacts of these events [30].

Seven indicators and fifty-seven risk factors related to water and heat waves were agreed upon through a meeting between key KMC stakeholders and the Sri Lankan Ministry of Environment. The probability of occurrence and impact of 57 risk factors were each measured on a five-point scale by 35 experts in Kurunegala city. The expert survey was conducted at the Consultant Workshop on 17–19 July 2019. Experts are water and heat wave experts who have been well aware of the current state of Kurunegala city for more than five years recommended by the KMC. The criteria for five-point scale are presented in Tables 1 and 2. Based on the average value of the experts for each indicator, the score was calculated by multiplying these values, and then each indicator and risk factor were ranked (Tables A1–A7).

Generally, adaptation plans should be established in various fields, including water, ecosystem, land use, agriculture, fisheries, and industry, etc. As cooperation from multiple ministries is needed, opinions from experts in other areas should also be collected. Therefore, a climate change awareness survey from stakeholders was conducted. They are 23 experts in the fields of environment, land use, health and forestry, etc., recommended by the KMC.

**Table 1.** Guidance on possibility of occurrence (Scale).

| Category | Score | Evaluation Criteria |
|---|---|---|
| | Very Severe (5) | Very likely to occur |
| | Severe (4) | Likely to occur |
| Possibility of Occurrence | Moderate (3) | May occur |
| | Low (2) | Low possibility to occur |
| | Very Low (1) | Not likely to occur |

**Table 2.** Guidance on impacts (Scale).

| Category | Score | Evaluation Criteria |
|---|---|---|
| | Very Severe (5) | • Severe and repeated damage to infrastructure and property<br>• Serious disruption in providing community services<br>• Overall impacts on vulnerable classes of the society<br>• Extensive damage to the ecosystem<br>• Influence the personal identity of individuals<br>• Detrimental influence on all parts and sectors of the society |
| | Severe (4) | • More severe than 'Moderate', yet less intense than 'Very Severe' |
| Impact (scale) | Moderate (3) | • Damage to infrastructure or property<br>• Expanded social Inequality<br>• Serious disruption in providing community services<br>• Mobilization of emergency services<br>• Damage of ecosystems<br>• Slight influence the personal identity of individuals<br>• Detrimental influence on all parts and sectors of the society |
| | Low (2) | • More severe than 'Very Low', yet less intense than 'Moderate' |
| | Very Low (1) | • Mild impacts<br>• Minor impacts on the economy<br>• Short-term, recoverable impact on the ecosystem<br>• Damage on not many personnel<br>• Tolerable damages |

A climate change awareness survey from women was also conducted, aiming to identify the impact of climate change on residents living there. In developing countries, the gender problem needs to be considered, and the damage caused by climate change can be more vulnerable to women and children [31,32]. The survey collected opinions from 40 women on the most serious climate-related problems in the city and the adaptation measures that the residents think are most needed. They are people of various occupations, including transportation, sales, self-employed, housewives, and students, and were conducted on-site surveys led by the KMC. We obtained their perspectives on the impact of climate change, as well as ideas on how to reduce this impact.

## 3. Results

### 3.1. Survey-Based Approach Assessment with Experts

The top 10 among the 57 risk factors of the seven indicators were analyzed. When all risk factors are compared, 'Lack of drinking water resources due to drought' is considered to be an exceptional risk factor (Table 3). When comparing DWR01 and WR01, the probability of DWR01 occurrence was lower than that of WR01, but its impact on the city was significant. This indicates that the lack of water is a serious problem; securing drinking water is the top priority of the climate change adaptation plan in Kurunegala city (Figure 1). The risk factors in TOP 10 are in DWR, WR, and HIH; therefore, we can observe that severe drought, water scarcity, and heat stress are the largest concerns for water and heat experts and policymakers of Kurunegala City (Table 3, Figure 2).

**Table 3.** Rank of risk factors.

| Code | Risk Factor | Avg. Possibility of Occurrence | Avg. Impact | Score | Rank |
|------|-------------|-------------------------------|-------------|-------|------|
| DWR01 | Lack of drinking water resources due to drought | 3.86 | 4.14 | 15.98 | 1 |
| WR01 | Lack of water for building maintenance and management | 4.03 | 3.91 | 15.76 | 2 |
| WM08 | Drying streams and water bodies (natural and artificial) due to drought | 3.94 | 3.83 | 15.09 | 3 |
| WR07 | Decrease in national water supply capacity due to the changes in rainfall pattern | 3.89 | 3.74 | 14.55 | 4 |
| HIH02 | Reduced function of green space and increased loss of green cover due to heat stress | 3.86 | 3.71 | 14.32 | 5 |
| WR10 | Increased irregularity in water supply among regions due to drought | 3.76 | 3.79 | 14.25 | 6 |
| WR05 | Increase in water demand due to the increase in crop evapotranspiration | 3.83 | 3.66 | 14.02 | 7 |
| WR06 | Increase in water demand for livestock and animal husbandry due to drought | 3.56 | 3.47 | 12.35 | 8 |
| WR09 | Uncontrolled use of groundwater due to lack of water | 3.6 | 3.43 | 12.35 | 9 |
| WM11 | Increased frequency of drought due to persistent non-precipitation days | 3.6 | 3.4 | 12.24 | 10 |

Moreover, its impact is of high concern to local experts in Kurunegala. Sri Lanka has been experiencing an ongoing drought since the beginning of 2016. In recent months, approximately one million people across the country have been affected by the drought [18]. Kurunegala city was specifically noted as the most damaged of the nine provincial Sri Lankan capital cities by drought between 1974 and 2017 [33]. Through interviews with residents, it was understood that as many as three to four months of severe drought each year have occurred in recent years.

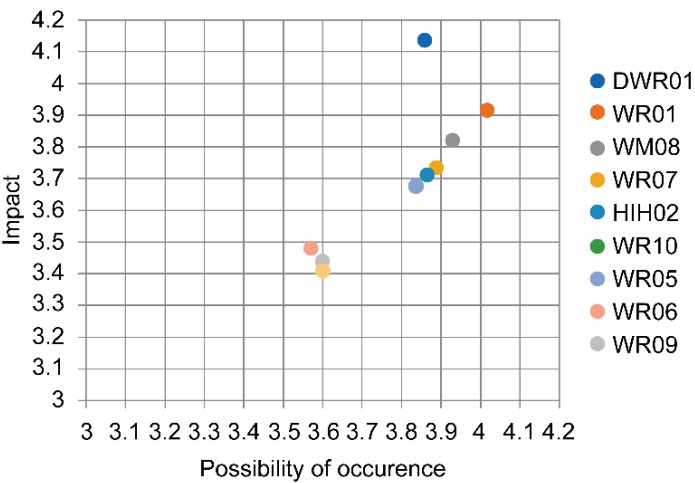

**Figure 2.** Top 10 risk factors.

As mentioned before, there is no indicator of drinking water resources in Korea. These differences relate to the basic management of drinking water and the securing of water resources. In addition to drought, there are also problems with water supply facilities and water treatment in developing countries (Table 4, Figure 3).

**Table 4.** Rank of DWR (drinking water resources) risk factors.

| Code | Risk Factor | Avg. Possibility of Occurrence | Avg. Impact | Score | Rank |
|---|---|---|---|---|---|
| DWR01 | Lack of drinking water resources due to drought | 3.86 | 4.14 | 15.98 | 1 |
| DWR02 | Disruption of drinking water facilities | 2.71 | 2.74 | 7.43 | 2 |
| DWR03 | Hindrance to water treatment efforts | 2.37 | 2.34 | 5.55 | 3 |

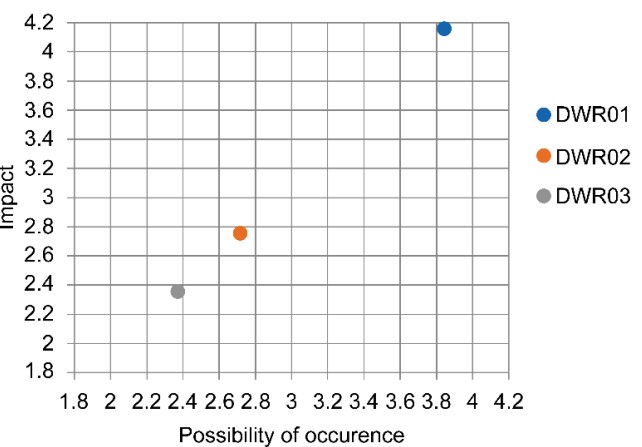

**Figure 3.** Risk to DWR (drinking water resources).

In the "Water resources" category, "Lack of water for building maintenance and management" and "Decrease in national water supply capacity due to rainfall pattern change" are high-effect factors, which indicates that Kurunegala city infrastructure requires the maintenance and local leaders to consider climate change as a high risk. In contrast, "Loss of fisheries because of the increase in harmful organisms (i.e., red tides and jellyfish) due to the increase in average water temperature" and

"Fluctuations in spawning habitat and season of fish, mollusks, and crustaceans due to the increase in water temperature" are low-effect indicators because Kurunegala is an inland region (Table 5, Figure 4).

**Table 5.** Rank of WR (water resources risk/vulnerability) risk factors.

| Code | Risk Factor | Avg. Possibility of Occurrence | Avg. Impact | Score | Rank |
|---|---|---|---|---|---|
| WR01 | Lack of water for building maintenance and management | 4.03 | 3.91 | 15.76 | 1 |
| WR02 | Change in the habitat of aquatic flora and fauna | 3.14 | 3.06 | 9.61 | 9 |
| WR03 | Loss of fisheries due to the increase in harmful organisms (e.g., red tide, jellyfish) due to average water temperature rising | 1.91 | 2.29 | 4.37 | 11 |
| WR04 | Fluctuations in spawning habitat and season of fish, mollusks, and crustaceans due to rising water temperature | 2.23 | 2.94 | 6.56 | 10 |
| WR05 | Increase in water demand due to increased crop evapotranspiration | 3.83 | 3.66 | 14.02 | 4 |
| WR06 | Increase in water demand for livestock and animal husbandry due to drought | 3.56 | 3.47 | 12.35 | 5 |
| WR07 | Decrease in national water supply capacity due to the changes in rainfall pattern | 3.89 | 3.74 | 14.55 | 2 |
| WR08 | Lack of water for SME industries due to drought | 3.37 | 3.31 | 11.15 | 8 |
| WR09 | Uncontrolled use of groundwater due to lack of water | 3.6 | 3.43 | 12.35 | 6 |
| WR10 | Increased irregularity in water supply among regions due to drought | 3.76 | 3.79 | 14.25 | 3 |
| WR11 | Change in groundwater level due to the increase in groundwater use | 3.46 | 3.31 | 11.45 | 7 |

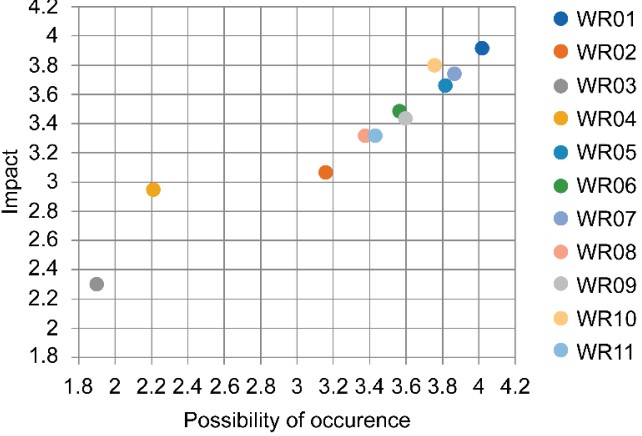

**Figure 4.** Risk of WR (water resources risk/vulnerability).

*3.2. Climate Change Awareness Survey*

The purpose of the climate change awareness survey was to understand the actual damage and impact of climate change that stakeholders and residents actually feel. To analyze the general population's climate change awareness, two surveys were conducted. (1) The "Stakeholder survey", was conducted to determine the views of general residents in Kurunegala. (2) The "Climate change awareness survey for women", was aimed at investigating differences in perceptions of climate change by women.

3.2.1. Climate Change Awareness Survey: Stakeholders

The Climate Change Awareness Survey was taken by 23 stakeholders who are all public workers in Kurunegala.

In terms of water, most respondents (44.9%) chose "Lack of drinking water" as the most serious climate change risk while "lack of living water" was the second (22.5%). This result shows that many respondents have a significant concern for the scarcity of drinking water. In contrast, the level of awareness among those who responded to the flooding and evacuation survey was relatively low (0%) (Figure 5, top).

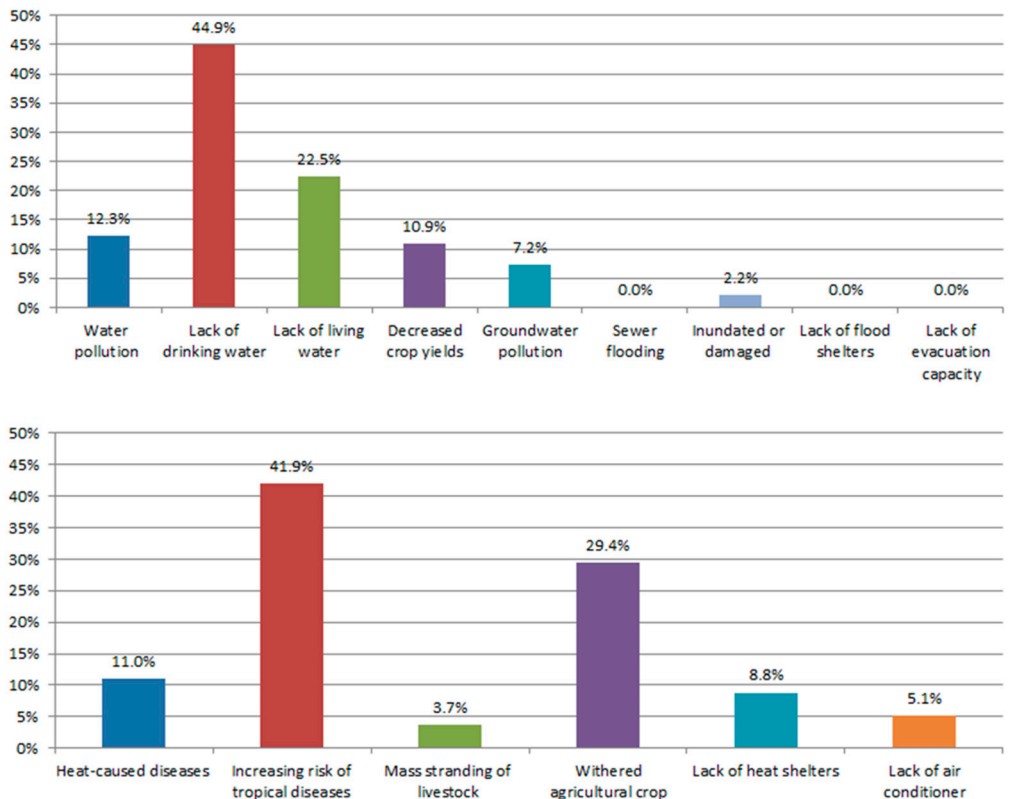

**Figure 5.** Most affected climate change risks in terms of water (**top**) and heat (**bottom**) as assessed by stakeholders.

Most respondents (41.9%) selected "Increasing risk of tropical diseases" as the most serious climate change risk associated with heat, followed by "Withered agricultural crop" (29.4%). However, as many of the respondents belong to the health department of KMC, they may have a particular interest in tropical diseases (Figure 5, bottom).

In terms of climate change measures for water, most respondents are concerned with the water supply, with "Supplying drinking water" (42.3%), "Supplying living water" (16.1%), and "Supplying agricultural water" (16.1%) as their three most critical concerns. People in Kurunegala city also ask for

increased attention to "Sewage treatment" (16.1%) while there are few demands for the "Installation of flood shelters" (0%) and "Supplying groundwater" (0.7%) (Figure 6, top).

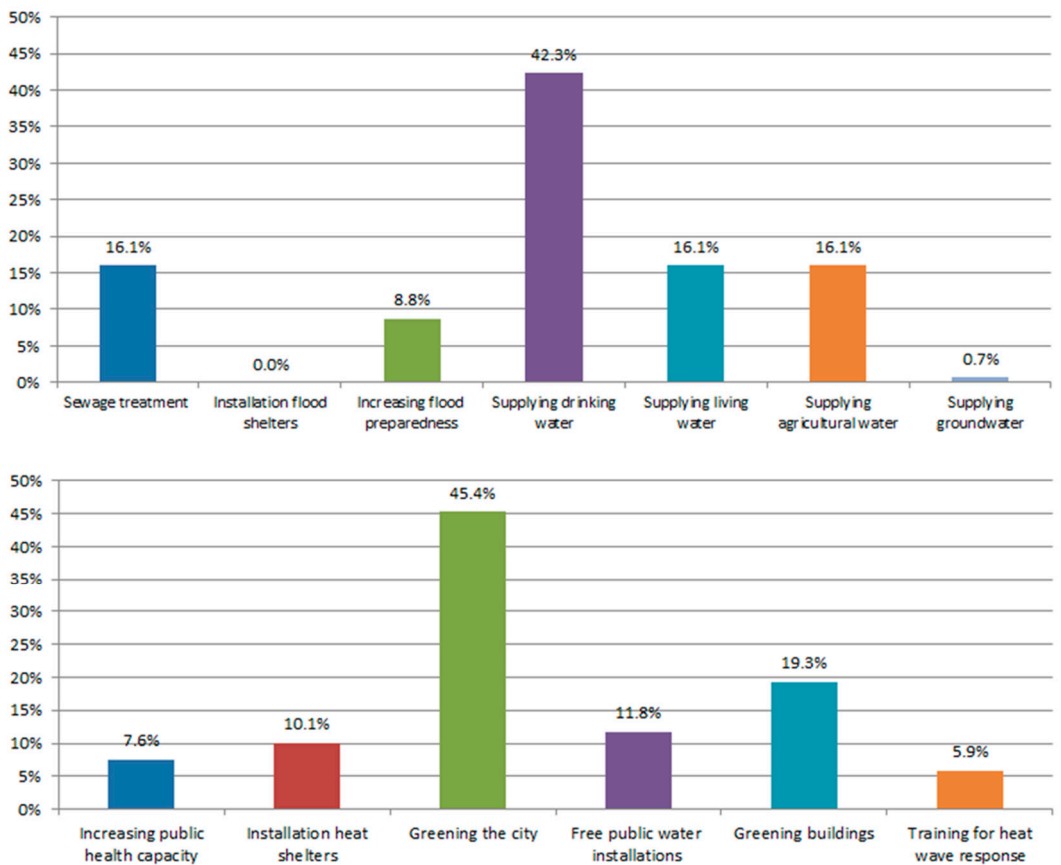

**Figure 6.** Most desirable climate change measures for the water (**top**) and heat (**bottom**) sectors as assessed by stakeholders.

More than half of the respondents chose "Greening the city" (45.4%) as the most desirable measure for addressing the heat issue, followed by "Greening buildings" (19.3%). This result shows that Kurunegala citizens want to have more green areas in their city (Figure 6, bottom).

3.2.2. Climate Change Awareness Survey: Women

We also analyzed climate change awareness from the female perspective. The "climate change awareness survey for women" was taken by 40 females in Kurunegala, aged 16–75.

Three main questions were utilized to analyze climate change awareness among women in Kurunegala city. The first question asked about the "level of climate change influence", where 62.5% of the respondents considered the climate change category as a high risk for their lives ("Very high": 37.5%, "High": 25%). Meanwhile, climate change is currently considered a huge risk; 72.5% of the respondents are concerned about climate change in the near future (Question 2, "Very high": 35%, "High": 37.5%). Moreover, most of the respondents identify the impacts of climate change as a severe risk to their society in Kurunegala. Almost half of the respondents, 47.5%, believe that the impacts of climate change are a "Very high" risk and 25% agreed that it is a "High" risk (Figure 7).

Similar to our stakeholder survey, most of the women selected "Lack of drinking water" (35.8%) as the most serious climate change risk, followed by "Lack of living water" (20.8%). In terms of heat, most women selected "Withered agricultural crop" (30%) as the most serious climate change risk while "Increasing risk of tropical diseases" (28.3%) (Figure 8) was their second most important concern.

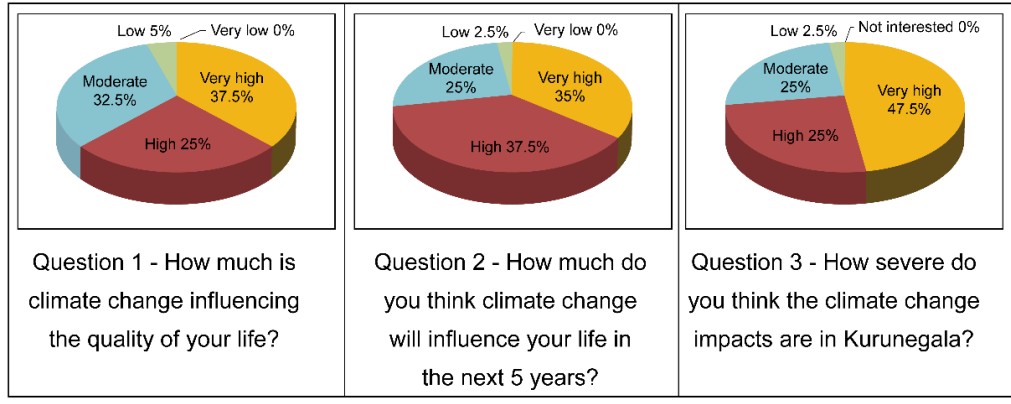

**Figure 7.** Women's awareness of local climate change impacts.

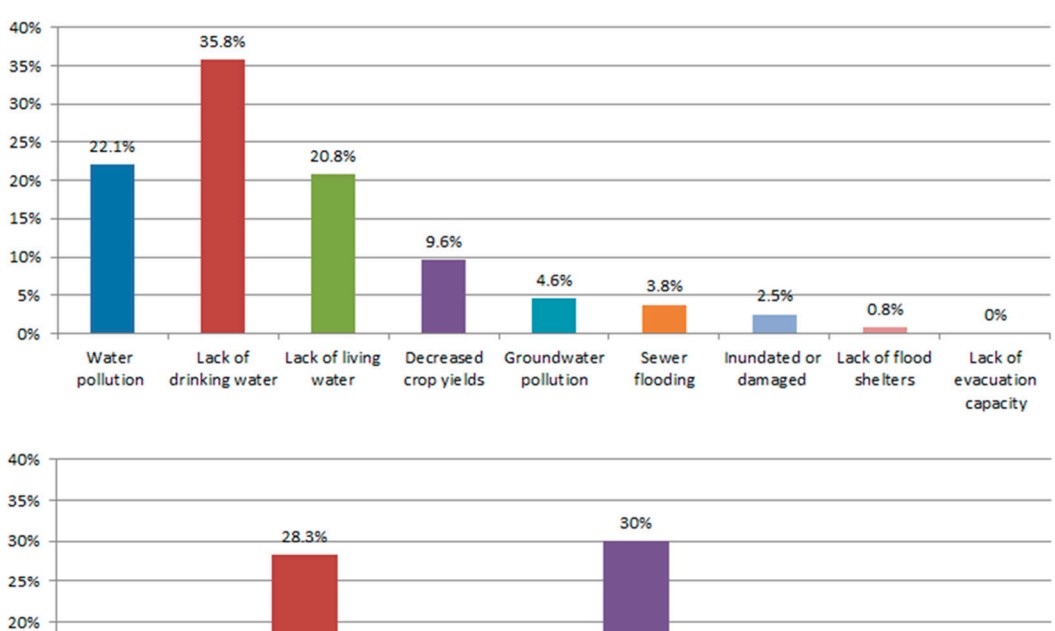

**Figure 8.** Most important climate change risks with respect to water (**top**) and heat (**bottom**) sectors as assessed by the women respondents.

The women surveyed viewed the water management policy priorities to be "Supplying drinking water" (33.8%), "Supplying living water" (20.8%), and "Supplying agricultural water" (15%). Women in Kurunegala also saw "Sewage treatment" (11.3%), "Installation of flood shelters" (6.3%), and "Supplying ground water" (5%) as important. Their responses to flood and underground water measures show that women have more realistic responses than the stakeholders (Figure 9, top).

Regarding heat, a large number of women (37.5%) chose "Greening the city" as the most desirable measure, followed by "Greening buildings" (18.8%) and "Free public water installations" (17.9%). In addition, "Increasing public health capacity" and "installation of heat shelters" have approval ratings of over 10% from the women respondents. This result shows that the women in Kurunegala city want to have more green areas in their city and public facilities to provide both water and heat protection (Figure 9, bottom).

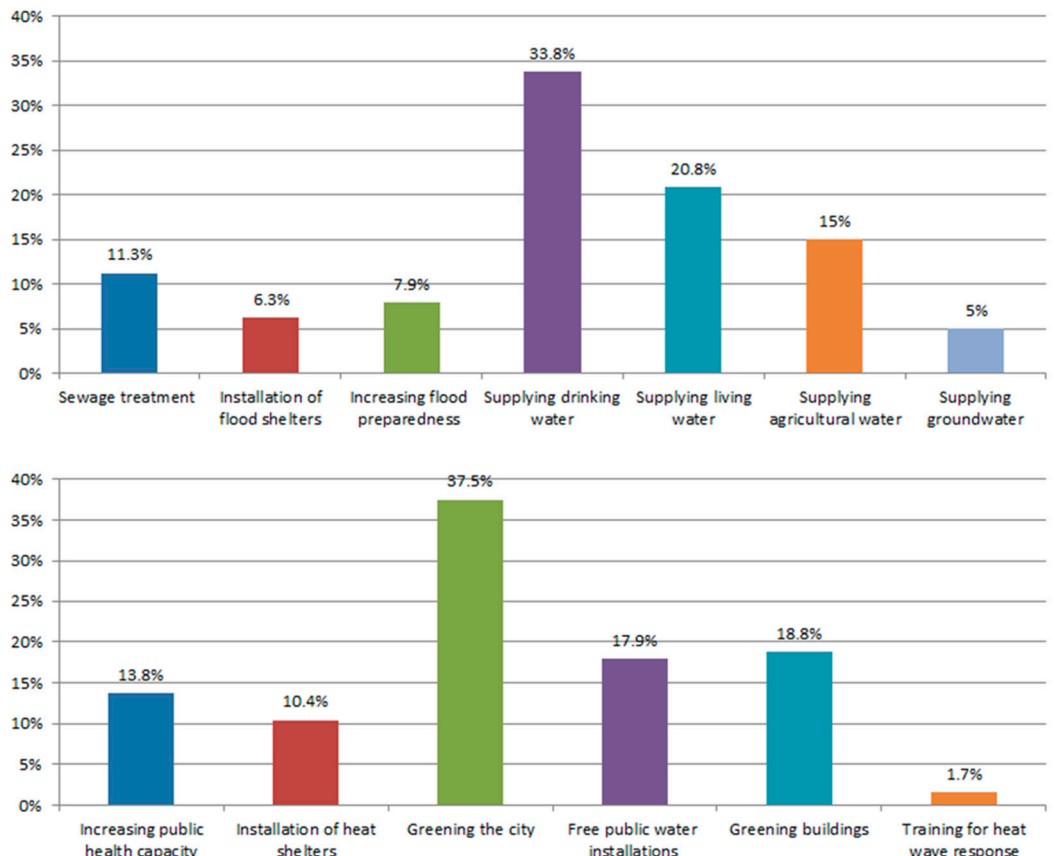

**Figure 9.** The most desirable climate change measures with respect to the water (**top**) and heat (**bottom**) sectors as assessed by the women respondents.

## 4. Discussion

The biggest limitation to implementing this risk assessment related to water and heat waves in Kurunegala city was the lack of quantitative data on their current status. This type of data is critically important for identifying the effects of climate change and establishing measures against it. These data are necessary for developing the ability to collect, organize, and interpret varied information that can identify and predict climate change.

There were several other limitations to conducting the risk assessment for Kurunegala city. The local government is aware of the problem; however, they are not aware of the processes required to analyze the situation and determine specific solutions. It is important to grasp the current problem and to accurately identify and analyze the current situation through both political and scientific methods.

Adapting to climate change is not something that one department can solve; it requires the recognition and cooperation of related departments and stakeholders as it requires joint work in various sectors of the ministry. Therefore, various stakeholders need to be aware of climate change and promote capacity building.

The process of listening to and gathering the opinions of residents is necessary for planning and establishing policies to protect what is considered vulnerable. When collecting opinions from stakeholders or women, it is necessary to gather opinions from as many diverse groups as possible. In this study, field surveys and stakeholder meetings were conducted to verify risks and establish adaptive measures based on the climate change risk assessment results. We engaged different stakeholders so that we could incorporate different opinions from water and heat experts, technical experts, and policymakers, as well as relevant authorities. As a result of our risk assessment of climate change, and the ongoing discussions through various stakeholder meetings, the most urgent issue appears to be securing drinking water during drought periods. In 2019, the drought period lasted

from June to August, but there was a rise in the ambient temperature beginning in April. In 2018, the dry period lasted only two months. The duration of hot, dry weather appears to extend each year.

The principal issue in water availability is that there is no reliable water source. The Deduru Oya reservoir had been utilized all of these years to provide paddy irrigation but the irrigation department built a weir in Deduru Oya to provide additional water for agricultural purposes. During the dry periods, the KMC and other related organizations have monthly discussions with the District Secretariat to assess the drought situation and the availability and distribution capacity of water sources. They desperately need to establish measures to manage water shortages during these increasing dry periods.

The other serious water availability issue is the high percentage of non-revenue water (NRW) consumption [34]. On a three-year average (2014–2016), the NRW is approximately 50% in Kurunegala [35]. The primary reason for the loss of NRW is frequent leaks within the aged pipe distribution network and defective domestic meters. As there is no digitized system for data collection and data storage at the KMC, it is necessary to formally design databases for continuous data collection and dissemination of information. The amount of bulk water bought from the Water Supply and Drainage Board, which categorizes (e.g., household, commercial) how the water units are being utilized, needs to be digitized. The distribution of wells, groundwater wells, and common bathing wells need to be mapped and monitored for whether they are in use or not during droughts and during normal capacity and use. Based on the data collected through an updated and modernized digitized system, it will be possible to determine which area has a leak and perform pipe network maintenance for that area.

Rainwater can be used to secure drinking water in the short term. Using the right efficiency membrane, drinking water can be secured in the current extreme situation. Water can be secured through large storage facilities in government facilities, such as city hall and school buildings, among others, or small storage containers in each household [36]. In the long term, better sustainable urban planning is needed in Kurunegala, such as the types now practiced at a global scale. The growing threat of climate change and environmental degradation has led to actions worldwide that have created resilient communities based on the principle of Natural Based Solution [37–39]. By applying new technologies, grey infrastructures are slowly being transformed to perform ecosystem services, aside from their conventional functions. Recent trends in urban design and planning involve the concept of water circulation cities, which involve the efficient utilization of all water resources by promoting natural water circulation and water reuse within the city area. For this to work, it is necessary to prepare proper water circulation measures in the city in advance.

Despite the fact that there are constant and increasing complaints of heat stress and related discomfort, this has not been fully or formally documented in Kurunegala. There are no factual reports connecting heat stress to health problems. We found that there is a slight increase in the incidence of dengue fever; however, no mortality statistics have been reported. There are several organizations involved with heat stress, but individual responsibilities are unclear. It is necessary to do physical and quantitative research on heat waves [40,41]. Based on the quantitative research, it needs to define the impact and damage status of heat stress and design databases for continuous data collection and dissemination of information to manage this factor. Due to the lack of a current database, it is necessary to manage the problem first, and then establish action plans for reducing the impact. In the short term, the installation of green and shade curtains in areas where damage is severe can reduce the damage from heat waves. Appropriate green-oriented landscape management mitigates the effects of heat waves [42,43]. Measures that can be formulated need to be implemented immediately.

Especially in the case of water shortage, the actual impact on residents can be substantial. As international climate-related funds have risen recently, it is also necessary to take advantage of them immediately to implement urgent policies and plans. Developing countries will need mid- to long-term support, rather than fragmentary and short-term support.

## 5. Conclusions

This study covered the process of risk assessment in Kurunegala city and its results. Risk assessment is an important component of the urban planning process to improve resilience against climate change. Among the various methods of climate change risk assessment, a survey-based assessment approach was conducted in line with the current status of Kurunegala city.

The climate change risk assessment was carried out through various stages. The major problems were determined to be water and heat waves after discussions with key stakeholders. Various impact indicators that could have a significant impact on water and heat waves were drawn up with KMC experts and stakeholders. Based on the water and heat wave indicators, two surveys were conducted with experts, stakeholders, residents, and local women to prioritize the risks that should be addressed. The climate change awareness survey is important in establishing plans for adaptation to climate change. This can help identify the experiences of local residents and what they actually need. The residents of Kurunegala city are highly concerned about climate change; however, most lack an awareness of water resources and health issues, including heat waves and tropical diseases.

As a result of the climate change risk assessment of experts and awareness survey of stakeholders and women, the major overall problem was determined to be the lack of drinking water. Water scarcity due to prolonged drought, which may be exacerbated by anthropological interventions, is also not properly documented. The problem of a lack of drinking water was found to be more serious due to the effects of severe drought caused by climate change, as well as a lack of water resource and water leak management. The impact of the heat wave is also understood to be significant. Residents have experienced heat stress in their communities, but there is no published information related to heat stress health hazards. No records are available from the Medical Officer of Health indicating the relationship between heat stress and health, such that it was not possible to fully understand the current situation. The climate change risk assessment results show that action plans are urgently needed to solve problems caused by the lack of drinking water and heat waves. Several adaptation measures are suggested: recirculating city water, rainwater utilization, effective use of existing water resources, including a reduced NRW rate, response to heat stress, data establishment for identifying and managing the impact of heat stress, and the installation of green and shade curtains.

These climate change risk outcomes play an important role in the development of climate change adaptation action plans. It is also important that the information being delivered to general residents about climate change and its associated risks is correct and clear. Policymakers in Kurunegala city should select and implement these policies in full consideration of the current situation, so the city will gradually grow into a climate-resilient city.

**Funding:** This study was conducted by the Korea Adaptation Center for Climate Change (KACCC) at the Korea Environment Institute (KEI) as part of the "*Sri Lanka, Technical support for climate smart cities* (2018-115)" project. This project is a pro-bono Technical Assistance project of the Climate Technology Centre Network (CTCN), entitled "*Developments of an urban adaptation plan for Kurunegala.*" This project was funded by the Korea Ministry of Science and ICT, "*Sri Lanka: Technical support for climate smart cities*" (grant number NRF-2018M1A2A2080813).

**Conflicts of Interest:** The author declares no conflict of interest.

## Appendix A  Risk Factors for Each Indicator for Survey-Based Approach Assessment and Experts Survey Result

**Table A1.** DWR risk factors and assessment result of Kurunegala.

| Code | Cause | Risk Factor | Experts Survey Result | | | |
|------|-------|-------------|---------|---|---|---|
| | | | **Possibility of Occurrence** | **Impact** | **Score** | **Rank** |
| DWR01 | Drought | Lack of drinking water resources due to drought | 3.86 | 4.14 | 15.9804 | 1 |
| DWR02 | Flood | Disruption of drinking water facilities | 2.71 | 2.74 | 7.4254 | 2 |
| DWR03 | Flood | Hindrance to water treatment efforts | 2.37 | 2.34 | 5.5458 | 3 |

**Table A2.** WM risk factors and assessment result of Kurunegala.

| Code | Cause | Risk Factor | Experts Survey Result | | | |
|------|-------|-------------|-------------------------|--------|-------|------|
| | | | **Possibility of Occurrence** | **Impact** | **Score** | **Rank** |
| WM01 | Flood | Disruption and functional degradation of river facility (e.g., embankment, bridge) | 2.23 | 2.26 | 5.0398 | 9 |
| WM02 | Flood | Influence on the operation of waterworks facilities (e.g., less access to operational mechanism) | 2.5 | 2.39 | 5.975 | 7 |
| WM03 | Flood | Economic loss due to impact from interruption of transportation and industry related with rivers and canals | 2.11 | 1.97 | 4.1567 | 11 |
| WM04 | Flood | Increase of damaged irrigation facilities due to flood | 2.23 | 2.17 | 4.8391 | 10 |
| WM05 | Flood | Increase of property assets loss due to increased frequency of flood | 2.48 | 2.33 | 5.7784 | 8 |
| WM06 | Flood | Damage to drainage facilities (e.g., storm sewer system) due to increase in urban sediment drainage and high volume of water | 2.77 | 2.46 | 6.8142 | 6 |
| WM07 | Drought/Flood | Increased cost risk due to lack of industrial water and water treatment due to water quality deterioration | 3.03 | 2.91 | 8.8173 | 3 |
| WM08 | Drought | Drying streams and water bodies (natural and artificial) due to drought | 3.94 | 3.83 | 15.0902 | 1 |
| WM09 | Flood | Decrease of safety and increase of destruction risk of water supply and repair facilities | 2.94 | 2.89 | 8.4966 | 4 |
| WM10 | Flood | Change in flooding occurrence and floodplains due to flood | 2.76 | 2.73 | 7.5348 | 5 |
| WM11 | Drought | Increased frequency of drought due to persistent non-precipitation days | 3.6 | 3.4 | 12.24 | 2 |

**Table A3.** WQA risk factors and assessment result of Kurunegala.

| Code | Cause | Risk Factor | Experts Survey Result | | | |
|---|---|---|---|---|---|---|
| | | | Possibility of Occurrence | Impact | Score | Rank |
| WQAE01 | Drought | Water quality deterioration due to pathogenic bacteria by temperature rising | 3.34 | 3.17 | 10.5878 | 4 |
| WQAE02 | Storm | Increase of water pollution risks due to excessive inflow of land pollution source (e.g., living sewage, industrial wastewater) by heavy rain | 3.11 | 3.2 | 9.952 | 5 |
| WQAE03 | Drought, Flood | Fluctuations in water ecology according to rainfall pattern change | 3.29 | 3.26 | 10.7254 | 3 |
| WQAE04 | Drought, Flood | Water quality deterioration due to rainfall pattern change | 3.41 | 3.29 | 11.2189 | 2 |
| WQAE05 | Ambient Temperature | Increase of algal blooms and deterioration of aquatic ecosystem due to average water temperature rising | 3.46 | 3.46 | 11.9716 | 1 |
| WQAE06 | Avg. water temp. rising | Increase of diseases and new pathogenic microorganisms occurrence due to water temperature rising | 3.11 | 3.14 | 9.7654 | 6 |
| WQAE07 | Avg. water temp. rising | Acceleration of extinction of endangered species and endemic species by climate change | 2.51 | 2.6 | 6.526 | 9 |
| WQAE08 | Storm | Effluence increase due to fertilizer, pesticides and animal wastes according to rainfall intensification | 2.89 | 2.94 | 8.4966 | 7 |
| WQAE09 | Temp. rising | Increase of abnormal reproduction of alien invasive species | 2.66 | 2.6 | 6.916 | 8 |
| WQAE10 | Avg. water temp. rising | Fluctuations in species composition and spawning season of fish | 2.53 | 2.52 | 6.3756 | 10 |
| WQAE11 | Avg. water temp. rising | Fluctuations in species composition and spawning season of mollusca and crustaceans | 2.4 | 2.4 | 5.76 | 11 |

**Table A4.** WR risk factors and assessment result of Kurunegala.

| Code | Cause | Risk Factor | Experts Survey Result | | | |
|------|-------|-------------|-----------------------|---|---|---|
| | | | Possibility of Occurrence | Impact | Score | Rank |
| WR01 | Heat stress, drought | Lack of water for building maintenance and management | 4.03 | 3.91 | 15.7573 | 1 |
| WR02 | Avg. water temp. rising | Change in habitat of aquatic flora and fauna | 3.14 | 3.06 | 9.6084 | 9 |
| WR03 | Avg. water temp. rising | Loss of fishery according to increase of harmful organisms (e.g., red tide, jellyfish) due to average water temperature rising | 1.91 | 2.29 | 4.3739 | 11 |
| WR04 | Avg. water temp. rising | Fluctuations in spawning habitat and season of fish and molluscs and crustaceans due to water temperature rising | 2.23 | 2.94 | 6.5562 | 10 |
| WR05 | Drought | Increase of water demand due to increase of crop evapotranspiration | 3.83 | 3.66 | 14.0178 | 4 |
| WR06 | Drought | Increase of water demand for livestock and animal husbandry due to drought | 3.56 | 3.47 | 12.3532 | 5 |
| WR07 | Drought | Decrease of national water supply capacity due to rainfall pattern change | 3.89 | 3.74 | 14.5486 | 2 |
| WR08 | Drought | Lack of water for SME industries due to drought | 3.37 | 3.31 | 11.1547 | 8 |
| WR09 | Drought | Un-controlled use of groundwater due to lack of water | 3.6 | 3.43 | 12.348 | 6 |
| WR10 | Drought | Increased gap of water supply among regions due to drought | 3.76 | 3.79 | 14.2504 | 3 |
| WR11 | Drought | Change in groundwater level due to increase of groundwater use | 3.46 | 3.31 | 11.4526 | 7 |

**Table A5.** SDF risk factors and assessment result of Kurunegala.

| Code | Cause | Risk Factor | Experts Survey Result | | | |
|------|-------|-------------|-----------------------|---|---|---|
| | | | Possibility of Occurrence | Impact | Score | Rank |
| SDF01 | Drought, flood | Increase of waterborne diseases | 3.11 | 3.09 | 9.6099 | 2 |
| SDF02 | Flood | Overflow of toilet | 2.83 | 2.69 | 7.6127 | 3 |
| SDF03 | Drought | Lack of water for toilet usage | 3.37 | 2.94 | 9.9078 | 1 |

**Table A6.** HF risk factors and assessment result of Kurunegala.

| Code | Cause | Risk Factor | Experts Survey Result | | | |
|------|-------|-------------|-----------------------|---|---|---|
| | | | Possibility of Occurrence | Impact | Score | Rank |
| HF01 | Flood | Increase of mortality rate due to disaster | 2.18 | 2.3 | 5.014 | 7 |
| HF02 | Flood | Increase of injury rate due to disaster | 2.29 | 2.5 | 5.725 | 5 |

**Table A6.** *Cont.*

| Code | Cause | Risk Factor | Experts Survey Result | | | |
|------|-------|-------------|-------------------------|--------|-------|------|
| | | | Possibility of Occurrence | Impact | Score | Rank |
| HF03 | Flood | Lack of Focused management on health vulnerable groups (e.g., infants and the elderly) | 2.74 | 2.66 | 7.2884 | 4 |
| HF04 | Flood | Increase of medical demand and lack of medical supply due to disaster | 2.66 | 2.82 | 7.5012 | 3 |
| HF05 | Flood | Increase of waterborne diseases (e.g., typhoid, cholera, bacterial heterogeneity) through water and food | 2.69 | 2.8 | 7.532 | 1.5 |
| HF06 | Flood | Increase of health problems (e.g., falls, trauma) due to safety accidents with flood | 2.31 | 2.34 | 5.4054 | 6 |
| HF07 | Flood | Increase of vector borne diseases | 2.69 | 2.8 | 7.532 | 1.5 |

**Table A7.** HIH risk factors and assessment result of Kurunegala.

| Code | Cause | Risk Factor | Experts Survey Result | | | |
|------|-------|-------------|-------------------------|--------|-------|------|
| | | | Possibility of Occurrence | Impact | Score | Rank |
| HIH01 | Avg. water temp. rising, drought | Increased intensification of the Urban Heat Island (UHI) phenomenon | 2.94 | 3.34 | 9.8196 | 4 |
| HIH02 | Heat stress | Reduced function of green space and increased loss of green cover due to heat stress | 3.86 | 3.71 | 14.3206 | 1 |
| HIH03 | Avg. water temp. rising | Increase of waterborne diseases such as diarrhea caused by drought | 2.8 | 2.86 | 8.008 | 10 |
| HIH04 | Avg. water temp. rising | Increase of morbidity and infectious diseases (e.g., infectious diseases) due to average temperature rising | 2.97 | 2.97 | 8.8209 | 8 |
| HIH05 | Heat stress | Increase of mortality due to heat wave | 2.77 | 2.69 | 7.4513 | 11 |
| HIH06 | Heat stress | Increase of cardiovascular diseases due to heat wave | 3.13 | 2.9 | 9.077 | 7 |

**Table A7.** *Cont.*

| Code | Cause | Risk Factor | Experts Survey Result | | | |
|------|-------|-------------|------------------------|---|---|---|
| | | | Possibility of Occurrence | Impact | Score | Rank |
| HIH07 | Heat stress | Increase of impacts on vulnerable groups due to intensification of Urban Heat Island (UHI) phenomenon caused by heat stress | 2.86 | 3.06 | 8.7516 | 9 |
| HIH08 | Heat stress | Decrease of labor productivity and labor time due to heat stress | 3.45 | 3.33 | 11.4885 | 3 |
| HIH09 | Heat stress | Increase of demand for consumer goods suitable for heat stress | 3.51 | 3.43 | 12.0393 | 2 |
| HIH10 | Heat stress | Increase of medical demand due to heat stress | 3.09 | 2.94 | 9.0846 | 6 |
| HIH11 | Heat stress | Increase of limited outdoor activities tourisms due to heat stress | 3.11 | 3.03 | 9.4233 | 5 |

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
