# Peer review of "Climate Change Risk Assessment for Kurunegala, Sri Lanka: Water and Heat Waves"

_climate, doi:10.3390/cli8120140_

Round 1

Reviewer 1 Report

Minor observations

Lines 214-217: Move the paragraph before figure 5.

Lines 258-263: Move the paragraph before figure 9.

Author Response

Thanks for your comments.

Lines 214-217 moved to line 223-226.

Lines 258-263 moved to line 272-277.

Reviewer 2 Report

This work has been properly redrafted. The issues raised were correctly presented. The author referred to all the comments from the previous version of the work and took them into account.

In my opinion this article can be accepted in the present form.

Author Response

Thanks for your comments.

Reviewer 3 Report

This paper is devoted to the evaluation of survey results concerning the impact of climate change to people who lives in Kurunegala in Sri Lanka According my opinion the results are interesting  and suitable for publication It will be good to compare the results from various cities in Sri Lanka or compare the results between  Korea and Sri Lanka.   Appendixes A and B should be put together, because it will be better to see the risk factor and its impact in one table.  

Author Response

Thanks for your comments.

Appendixes A and B were combined (line 407)

Reviewer 4 Report

Review of the manuscript entitled: 

Climate Change Risk Assessment for Kurunegala, Sri Lanka: Water and Heat Waves

             by

Hanna Cho

The author carried out the climate change risk assessment through various stages. The author believes that the availability of water and heat waves are big problems in Sri Lanka and, to better understand and mitigate them,  formulates different qualitative indicators of drought and heat waves. Based on the water and heat wave indicators, the author conducts two surveys  with experts, stakeholders, residents, and local women to prioritize the risks that should be addressed. The author  notes that there is  a large lack in  awareness of water resources and health issues and it is difficult to quantitatively express the risks of climate change. The author  attempts to evaluate climate change with as much objectivity as soon as possible.

Criticism:

The author analyses  the increase in droughts and heat waves too qualitatively and use indices that do not take into account the numerous physical indices characterizing  the drought and heat wave risk. Owing to this qualitatively analysis, the author must introduce more quantitative and physical parameters such as ENSO index which is considered  responsible for droughts and heat waves.

Moreover, the author must take into account the long cycle behaviour of climatic change related to solar activity such as the 60-yr cycle.

The climate of Kurunegala (see rows 66-71) is reported too succinctly. It must be more detailed and more explained especially regarding the reliability of the data used.

Minor points:

row 59  is typically defines  must be corrected     is typically defined

Anyway, I find the paper a good first contribution to the characterization of water and heat wave risk  in Sri Lanka. In other successive paper, the author will have to face the same phenomena of drought and heat waves from a more physical and quantitative point of view.

I suggest  the insertion of the following paper:

1)Di Cristo R., Mazzarella A., Viola R.: An analysis of heat index over Naples (Southern Italy) in the context of European heat wave of 2003, Natural Hazards, DOI: 10.107/s11069 -006-0033-7, 40, 373-379, 2007.

Author Response

Thanks for your comments

The need for quantitative assessment was emphasized(lines 95-96, 351) and the reference was added(lines550-556). I will refer to the specific method when conducting a quantitative assessment in the future.

I described the climate of Kurunegala in more detail(lines 67-82)

Round 2

Reviewer 4 Report

The authors have accepted all my sugegstions

This manuscript is a resubmission of an earlier submission. The following is a list of the peer review reports and author responses from that submission.

Round 1

Reviewer 1 Report

The journal of Climate aims at the climate processes of the earth, covering all scales and involving modelling and observation methods. Although climate mitigation and adaption were included as subjects of journal scopes, this paper adopted a survey-based approach to study climate risk assessments of the Kurunegala city in Sri Lanka based on 57 risk factors selected from 84 risk factors developed by a Korean study.

Without any scientific data, results of risk assessments will be very difficult to be useful for the planning of adaption measures, as well as whether risk awareness is clear enough to be supported by scientific data.

A survey-based approach to determine climate change risk assessment can be a subset to support scientific-based and quantitative assessments, but as a whole for risk assessments is not acceptable, not to mention to be published in a scientific journal. Another concern is the survey-based approach normally contains may social factors during the process of survey which will affect the result of survey, which were not well presented in the paper. I would like to suggest the author should find another suitable journal for their survey-based approach.

The following comments should be considered for the author in future submission,

  1. Provide some references using a similar approach and successful in planning on adaption measures.
  2. The survey was conducted in which year? Took how long?
  3. The survey was conducted with 35 climate change adaption experts. This is very unclear. How the 35 experts selected in terms of professional experiences in climate adaption?
  4. Why give a separate survey on women, but following the same survey-based approach? So there is no woman include in the 23 stakeholders survey? The author should highlight why need to focus on women survey for the area.
  5. How the 23 stakeholders of public workers selected for the survey study. What are the daily works for those 23 public workers, outside or indoor?
  6. After the review, there are 57 risk factors adopted from the original 84 frisk factors developed in a Korean study (no references). Who took the review? Who decides the final 57 factors were suitable for the study area.
  7. There is no clear picture of the final 57 factors that looks like. Only the most important factors presented with figures and tables. How about those with less attention during the survey? Are they really less important or simply lacking data to support their importance?
  8. Wording selections should be improved. For examples, use “challenges” to replace “problems” in Line 8; remove “implemented and” in Line 12; “global warming” does not represent the whole of “climate change” in Line 19 (so global warming or climate change related risks will be the focus of this study?), etc.
  9. Line 45: improve the quality of four small figures in Fig. 1. and provide explanations in the text and necessary legends of years to distinguish their differences

Reviewer 2 Report

Minor observations

Too few references from indexed journals.

Line 128: “… after which indicators …” which is probably each.

Paragraph 3.1. Lines 135 to 176: It is interesting to develop a bit the explanation related on how values from Tables 1 to 4 were obtained / established, and also about the graphical representations from fig. 2 to 5, Impact and Possibility of occurrence.

Table 1 were obtained. Also, I feel the need that the author comment about

Line 194: Figure 6. I suggest using for both graphs the same maximum for vertical axis, 50%. Why to use decimals on vertical axis labels, e.g. 30.0% and not 30%?

Line 209: Figure 7. Same observations as for Line 194: Figure 6.

Line 229: Figure 9. Same observations as for Line 194: Figure 6; using for both graphs the same maximum for vertical axis, 40%.

Line 238: Figure 10. Same observations as for Line 229: Figure 9. Here, no observation regarding using the same maximum for vertical axis, 40%.

Reviewer 3 Report

In the considered work the author discuss important and interesting problem - climate change risk assessment for Kurunegala, Sri Lanka. In this work we can find a sentence: "The biggest limitation to implementing this risk assessment related to water and heat waves in Kurunegala city was the lack of quantitative data on their current status" (Discussion). Therefore, the author presented some cosiderations based on data collected during discussions with local people:
"Thus, a climate change risk assessment was implemented and conducted for both the water and heat wave risks via discussions with key stakeholders" (Introduction).

In my opinion, this work does not contain solution of any scientific problem. Moreover, any scientific problem have not been considered.

Therefore, in my opinion, the presented work can be used as a brochure on the problems of local people.

Finally, a major revision of this work should be suggested.

Reviewer 4 Report

This work deals with a risk assessment focused on water and heat waves in Kurunegala city, Sri Lanka, by means of a qualitative methodology, based on surveys and stakeholders engagement. This could be a valuable work since it shades lights on climate change risks and impacts in a stakeholder-oriented strategy. I found particularly interesting the importance attributed to the women point-of-view on the perception of climate change impacts. Nevertheless, I think that additional and significant work should be done before publication, thus major revisions are required. The introduction provides a good description of Sri Lanka and the city context but general literature on the topic is lacking and needs to be improved. The methodology has not been sufficiently detailed and explained and, in my opinion, should be considerably broadened. Results and Discussion sections are consistent with the findings. More details are provided hereinafter.

Introduction

The authors should contextualized the climate change risk assessment on a broader scale, relying on the existing literature on this topic and afterward focus on the national context.

L 19-20 Please rephrase

L 29-30 This statement is pretty strong, on what basis can we state that Sri Lanka is one of the most affected countries by climate change in 2018? Please add more details. 

L 48-56 This paragraph describes the study area. It should be moved to another section (e.g., Material and Methods)

L 95 Reference? 

L 107 "after the review..." please add details.

L 108 It is not clear how did you evaluated the risks, what are the assumptions behind that and the methodology. This part needs to be considerably improved and further details should be provided.

L 126-127 How did you measure it? How did you get this score?